# High Energy Density in All-Organic Polyimide-Based Composite Film by Doping of Polyvinylidene Fluoride-Based Relaxor Ferroelectrics

**DOI:** 10.3390/polym16081138

**Published:** 2024-04-18

**Authors:** Chengwei Wang, Yue Shen, Xiaodan Cao, Xin Zheng, Kailiang Ren

**Affiliations:** 1Beijing Institute of Nanoenergy and Nanosystems, Chinese Academy of Sciences, Beijing 100083, China; wangchengwei@binn.cas.cn; 2Center on Nano Energy Research, School of Physical Science and Technology, Guangxi University, Nanning 530004, China; shenyue08091998@163.com (Y.S.); caoxiaodan@binn.cas.cn (X.C.); zhengxin9911@gmail.com (X.Z.); 3School of Nanoscience and Engineering, University of Chinese Academy of Sciences, Beijing 100049, China

**Keywords:** all-organic composites, high energy density, PVDF-based relaxor ferroelectrics, high-temperature capacitors, polyimide film

## Abstract

Recently, due to the advantages of superior compatibility, fewer interface defects, and a high electric breakdown field, all-organic dielectric composites have attracted significant research interest. In this investigation, we produced all-organic P(VDF-TrFE-CFE) terpolymer/PI (terp/PI) composite films by incorporating a small amount of terpolymer into PI substrates for high energy density capacitor applications. The resulting terp/PI-5 (5% terpolymer) composite films exhibit a permittivity of 3.81 at 1 kHz, which is 18.7% greater than that of pristine PI (3.21). Furthermore, the terp/PI-5 film exhibited the highest energy density (9.67 J/cm^3^) and a relatively high charge–discharge efficiency (84.7%) among the terp/PI composite films. The energy density of the terp/PI-5 film was increased by 59.8% compared to that of the pristine PI film. The TSDC results and band structure analysis revealed the presence of deeper traps in the terp/PI composites, contributing to the suppression of leakage current and improved charge–discharge efficiency. Furthermore, durability tests confirm the stability of the composite films under extended high-temperature exposure and cycling, establishing their viability for practical applications.

## 1. Introduction

With the development of electronic systems, increasing new requirements for high-performance dielectric capacitors has become critical [1,2]. Compared to ceramic capacitors, polymer capacitors have attracted extensive research attention due to their advantages of low cost, high electric breakdown field, and ease of fabrication [3]. In high-temperature working environments, such as electric cars and aerospace applications, polymers with high glass transition temperatures (T_g_), including polyimide (PI) [4], poly(ether-ether-ketone) (PEEK) [5], polyetherimide (PEI) [6], polyaniline (PANI) [7], and poly(ether-ketone-ketone) (PEKK) [8], are greatly utilized. Among these high-T_g_ polymers, due to its advantages of high solubility in solvents and ease of processing, PI has been selected as a great candidate for high-temperature capacitor applications. As a representative high-temperature polymer, polyimide (PI) exhibits exceptional glass transition temperatures (T_g_) exceeding 360 °C.

However, the rigid conjugated structure on the main carbon chain restricts the motion of dipoles in high-T_g_ polymers, resulting in relatively low dielectric constants in these polymers. Moreover, due to π-π stacking between conjugated structures in high-Tg polymers, electron transport is prone to occur at elevated temperatures, which leads to relatively high conduction loss [9]. To overcome these issues, composite materials incorporating other dopers to enhance the breakdown field and the energy storage performance of PI polymers have become a cost-effective strategy. For example, incorporating highly dielectric ceramic nanoparticles into a polymer matrix can effectively improve the permittivity of composite films. However, due to the mismatch in the dielectric constants, electric field distortions occur at the filler–polymer interface, leading to high dielectric losses and leakage currents, which cause a reduced electric breakdown field in the composites [10]. Previously, it was demonstrated that the leakage current resulting from dielectric mismatch can be effectively reduced through the surface modification of ceramic fillers, such as PMMA or polydopamine [10,11,12,13,14,15,16]. However, the complex coating process on ceramic fillers is not conducive to mass production [10,11,12,13]. In addition, the dielectric properties of composite films can also be effectively improved by adding wide-bandgap nanofillers, such as boron nitride nanosheets (BNNSs) [14,15], nanodiamonds [16], Al_2_O_3_ [17,18,19], and MgO [20,21]. However, the poor compatibility between inorganic fillers and polymers leads to large dispersion and agglomeration, resulting in degradation of the electric breakdown field, which will eventually reduce the energy density of composite films.

Recently, all-organic dielectric composites have attracted great research interest for high-energy density capacitors because of their superior compatibility, fewer interface defects, and high electric breakdown field. In 2019, Zhang et al. mixed PEMEU (poly(ether methyl ether urea)) polymer with PEI to fabricate PEMEU/PEI films using a co-pouring method. With the disruption of hydrogen bonding among PEMEU, the composite film exhibits increased free volumes and enhanced dipole motions, leading to a higher dielectric constant of PEMEU/PEI [22]. In 2021, Zhang et al. reported PI/PEI blended films using a simple solution casting method for high-energy capacitor applications [23]. Due to the different electrical properties of the conjugated structure in the blended films, many defects are eliminated and the free volume is reduced by electrostatic force stacking between the two molecular chains. Compared to pristine PI and PEI, the blend film exhibited an improved breakdown strength of 550 MV/m, up to 200 °C. Similar phenomena were observed in other all-organic blend films, such as PI/PEEU, by Zhang et al. [24]. In 2020, Yuan et al. reported an all-organic hybrid composite consisting of polyether imide (PEI) and a high electron affinity molecular semiconductor [25]. A molecular semiconductor may form electron traps to capture free electrons, which inhibits leakage current and increases the electric breakdown field and energy density of the composite films. However, the slightly large percentage of semiconductor molecules may increase the leakage current of the composite. However, due to the linear dielectric behavior of the PI and PEI polymers, previous studies have shown limited improvements in both the dielectric constant and electric breakdown field. Hence, all organic polymer composites with relatively high dielectric constant and energy density still need to be studied for high-temperature capacitor applications.

Relaxor ferroelectric terpolymers P(VDF-TrFE-CFE) (poly(vinylidene fluoride-trifluoroethylene-chlorofluoroethylene)) exhibit a relatively large dielectric constant of ~60 and a small remnant polarization (Pr). However, due to the relatively low T_g_ and breakdown field (<350 MV/m) of terpolymer films, their use in high-energy density capacitors is limited, especially at high temperatures [26,27,28,29,30]. Previous research has shown that by introducing polar nanoregions (PNRs) in relaxor ferroelectrics, relaxor ferroelectrics can exhibit a greatly improved saturation polarization (P_s_) and an enhanced energy density [26]. In this investigation, a small amount of terpolymer was incorporated into PI substrates to fabricate an all-organic terpolymer/PI (terp/PI) composite film using a solution casting method. Then, we systematically analyzed the microstructure, dielectric constant, and energy storage properties of the terp/PI composite films. The incorporation of a small amount of P(VDF-TrFE-CFE) in the PI substrate can enhance both the dielectric constant and breakdown strength of the terp/PI films. In addition, the introduction of a terpolymer can produce depth traps, which inhibit leakage current, and improve the breakdown electric field of the composite film. Terp/PI composites show great promise for developing high-temperature capacitor materials. 

## 2. Experimental Section

### 2.1. Materials

As shown in Figure 1a, the composite films were prepared using a solution casting method. First, 1.5 g of polyamide acid solution (PAA) (15 wt.%, Saint Marvel, Run Chuan Plastic Materials Co., Changzhou, China) was added to 7.0 mL of N’N-dimethylacetamide (DMF) solvent in a clear glass bottle and was stirred for 4 h at room temperature until it was completely dissolved. Next, based on the mass of PAA powders, 1 wt.%, 5 wt.%, and 10 wt.% P(VDF-TrFE-CFE) powders (59.2/33.6/7.2 mol%, Piezo Tech. Inc., Sarthe, France) were weighed and added to the obtained PAA solution. The mixture was sonicated for 2 h in a water bath at room temperature and then magnetically stirred for an additional 4 h until it was completely dissolved. Next, the terp/PAA solution was cast on a clear glass plate and dried at 60 °C for 24 h. The obtained terp/PAA film was heated at 200 °C for 12 h to remove the residual solvent and moisture. This step is also beneficial to the thermalimide process of the PAA film. Afterwards, the terp/PAA film was heated at 250 °C and 300 °C for 2 h to continue the thermalimide process and obtain the terp/PI film. Finally, the obtained terp/PI films were peeled off in deionized (DI) water from the glass plate and vacuum dried at 60 °C for 24 h. In this investigation, terp/PI composite films with various terpolymer concentrations of 1%, 5%, and 10% were named terp/PI-1, terp/PI-5, and terp/PI-10, respectively. Similarly, pristine PI and terpolymer films were prepared using the solution casting method.

### 2.2. Characterization of Materials

The surface morphology of the cross-section and the surface of the fabricated composite film was observed using scanning electron microscopy (SEM) (SU 8020, Hitachi High-Tech, Tokyo, Japan), equipped with energy-dispersive spectroscopy (EDS). Transmission electron microscopy (TEM) images of the composite films were obtained using an FEI Tecnai G2 F20 (Thermal Fisher Scientific Inc., Waltham MA, USA). The X-ray diffraction (XRD) patterns of the terp/PI films were collected using a Cu Kα source (40 kV, 40 mA) with a scan step size of 0.013° (PANalytical Ltd., Almelo, The Netherlands). Dynamic mechanical analysis (DMA) was conducted using a DMA Q800 instrument (TA Instruments, Newcastle, DE, USA) in a nitrogen environment, with a heating rate of 5 °C/min. Thermogravimetric analysis (TGA) and differential scanning calorimetry (DSC) were performed using a TGA/DSC1 (Mettler-Toledo LLC., Columbus, OH, USA) with a heating rate of 10 °C/min. Fourier transform infrared (FTIR) spectroscopy was performed using a Bruker Vertex80V instrument (Bruker Corp., Billerica, MA, USA). The ultraviolet photoelectron spectroscopy (UPS) data were obtained using a PHI5000 Versa Probe III (ULVAC-PHI, Kanagawa, Japan), with an incident photon of 21.2 eV. A UV–visible light–infrared (UV–VIS–IR) spectrophotometer was used to measure the absorbance, using a UV3600 spectrophotometer (Shimadzu, Kyoto, Japan).

For the dielectric measurements, gold electrodes, 3 mm in diameter, were sputter coated on both sides of the terp/PI films. The dielectric constant and dielectric loss of the terp/PI films were measured using an LCR meter (Keysight 4980A, Santa Rosa CA, USA) from 1 kHz to 1 MHz. The electric field–polarization (E–P) loops of the terp/PI films were measured at 100 Hz using a Radiant Precision Multiferroic System (Radiant Technologies Inc., Albuquerque, NM, USA) at both room temperature and 150 °C. The thermally stimulated depolarization current (TSDC) was measured using a Keithley 6514 electrometer (Tektronix, Beaverton, OR, USA) and a temperature control test system (Poly k EC1A, Poly K Technologies, PA, USA) from room temperature to 250 °C. Before the TSDC measurement, the film was polarized under a DC electric field of 40 MV/m at 150 °C for 1 h. During the TSDC measurement, the film was slowly heated from 30 °C to 250 °C at a rate of 2 °C/min to record the change in the depolarization current as a function of temperature. The high-temperature durability of the terp/PI composite films was measured via the cyclic charge–discharge of the samples using a Radiant precision multiferroic system under 200 MV/m at 150 °C.

In addition, Multiwfn 3.8 software was used to compute the distribution of the electrostatic potential (ESP) within different ranges of electric potentials at the same level, and several statistical parameters related to the ESP were used in the calculation. Subsequently, color-filled iso-surface graphs for the molecular electrostatic potential were generated using a visual molecular dynamics (VMD) program.

## 3. Results and Discussion

As shown in the SEM images in Figure 1b, the terpolymer was uniformly dispersed in the PI substrate without obvious phase separation or aggregation. Subsequently, the distribution of elements in the terp/PI composite film was analyzed using EDS (energy-dispersive X-ray spectrometry). Appendix A shows that N, F, and O are uniformly distributed. For the TEM image, the sample was prepared by depositing a diluted solution of terpolymer/PAA on a copper grid followed by drying and thermal annealing in an oven. High-resolution TEM images (Figure 1c) revealed crystalized regions, in which distinct lattice patterns of the terpolymer coexisted with the amorphous PI, and the EDS pattern indicated the presence of polycrystalline terpolymer in the composite film. The images of the fabricated terp/PI-5 film are shown in Appendix A. Furthermore, the terp/PI-10 composite film exhibited a noticeable phase separation phenomenon with increasing terpolymer concentration, as depicted in Appendix A. According to previous studies, the phase separation phenomenon may arise from the solubility difference caused by the structural variations between the two types of molecular chains, ultimately resulting in a pronounced phase separation induced by high contents of fillers [26]. Normally, terpolymer molecules tend to form dendritic structures through aggregation during the PAA thermal imidization process [26]. Therefore, excessive terpolymer filler content may lead to phase separation and reduce the breakdown field of terp/PI composites. Therefore, a percentage of less than 10% was employed in the PI substrates to make the terp/PI composites.

Furthermore, the XRD analysis revealed that the spectrum of the pristine PI film exhibited broad amorphous peaks, while the spectrum of the composite film showed a characteristic peak of the α phase for the terpolymer at 18.2°, indicating the successful incorporation of the terpolymer into the PI matrix (Figure 2a) [27,28,29]. The intensity of the characteristic peak at 18.2° increases with increasing terpolymer content in the composite films. The flexibility of the terpolymer chain may additionally reduce the free volume and result in the tighter packing of the PI chains, causing a shift toward higher angles for the amorphous peak of the PI matrix in the XRD pattern [30]. The average chain spacings of the composite films with different terpolymer contents were calculated by employing Bragg’s law, and the results are outlined in Appendix A. From the table, the average spaces of the Terp/PI composites decrease with increasing terpolymer content.

The FTIR spectra of the terp/PI composite films are shown in Figure 2b. The data demonstrate that there were no obvious absorption peaks at approximately 1660 cm^−1^ and 1560 cm^−1^ in either the PI or terp/PI composite films, indicating that the imidization reaction was complete in the PI film [31]. The absorption peaks at 719 cm^−1^, 1712 cm^−1^, and 1776 cm^−1^ correspond to the bending vibration and the symmetric and asymmetric stretching vibrations of the C=O bond of the imide groups [32], respectively. Furthermore, as shown in Appendix A, the terp/PI composite films exhibit a distinct absorption peak at 780 cm^−1^, which is attributed to the TGTG’ conformation of the α-phase of the terpolymer [28]. In addition, the absorption peak of T_3_G in the terp/PI composite film at 513 cm^−1^ moves toward a higher wavenumber [33,34,35,36]. In summary, the FTIR spectra confirmed the absence of any chemical bond formation between the PI film and the terpolymer film, indicating that the terpolymer was present in the terp/PI film through physical blending.

The resulting mechanical properties of the terp/PI films are shown in Figure 3e. The terp/PI-5 film exhibited the highest Young’s modulus herein (3.33 GPa). This is attributed to the reduction in the free volume, which may tighten the packing between the molecular chains and increase the strength of the material. However, as the terpolymer content increased to 10 wt.%, the Young’s modulus decreased to 2.62 GPa. This may result from the weak Young’s modulus of the terpolymer, and the large percentage of low Young’s modulus materials can reduce the Young’s modulus of the composite films. The DMA curve storage loss data are commonly used to characterize the glass transition temperature (T_g_) of polymer materials. As shown in Figure 3f, a relaxor loss peak exists near 110 °C for the terp/PI composite film, which is attributed to the increased mobility of the terpolymer. However, the highest relaxor loss is still less than 5% for terp/PI-10, which is still within a reasonable range [29,33]. The inset in Figure 3f shows an enlarged view of the mechanical loss peak. Compared to that of the pristine PI film, the T_g_ of the terp/PI composite film first increases and then decreases, while that of the Terp/PI-5 composite film shows the highest T_g_ of 412.8 °C, which is consistent with the Young’s modulus values in Appendix A. The high Young’s modulus in the terp/PI composites may significantly reduce the free path of electrons, which can significantly increase the breakdown field of the material. Therefore, the terp/PI-5 film was focused on to investigate high-energy density capacitor applications in this study.

The thermal stability of the terp/PI composite films was subsequently evaluated using TGA. As shown in Appendix A, the onset decomposition temperature of the film decreases with increasing terpolymer content. This can be attributed to the much lower decomposition temperature of the terpolymer film (Appendix A). The thermogravimetry results were quantitatively analyzed using two parameters—a temperature of 10% weight loss and a residual mass ratio at 700 °C (Appendix A). These results indicate that the terp/PI-5 film exhibits the highest thermal stability and mechanical strength, which is conducive to avoiding electrical–mechanical breakdown. Therefore, terp/PI-5 is an ideal candidate for applications in high-temperature capacitors.

Afterward, the dielectric properties of the terp/PI composite films were investigated. As shown in Figure 2c, the permittivities of the terp/PI-1, terp/PI-5, and terp/PI-10 films reached 3.51, 3.81, and 4.27, respectively, at 1 kHz, which are 9.3%, 18.7%, and 33% greater than that of pristine PI (3.21). From previous publications, it is known that the dielectric constant of the composite is proportional to the dielectric constants of the substrate and the dopant [37]. Therefore, terp/PI-10 exhibits the highest dielectric constant of all the terp/PI composites (Figure 2c). However, the molecular chain structure of terpolymer differs significantly from that of PI, and their physical mixing can result in phase separation between them when there is a high content of terpolymer. This may reduce the breakdown field of terp/PI composites with a high content of terpolymer. Therefore, our focus was on the terp/PI-5 composite film for the study of high-energy density capacitors.

For the composite film, the dielectric loss became more severe with increasing terpolymer content. However, the dielectric loss of the terp/PI-10 composite film is ~3.5% at 1 MHz, which is slightly higher than that of the terp/PI-5 composite. During this measurement, a 200 g metal piece was placed on top of the holder of an HP LCR meter to ensure good contact between the holder and the composite sample. According to the literature, the high dielectric loss in the terp/PI-10 sample could be caused by the dipole relaxation in the terpolymer [38]. In addition, the data also indicate that the dielectric loss was slightly high at low frequencies, which could be due to the inconsistency of the HP LCR meter at low frequencies (<1 kHz). 

Furthermore, the polarization–electric field (P-E) loops of the pristine PI and terp/PI composite films were measured at room temperature; the data are shown in Figure 3a. With increasing terpolymer content, the saturation polarization (Ps) of the terp/PI composites gradually increased, which is consistent with the dielectric constant. The P_s_ of the terp/PI-5 and terp/PI-10 are 44.3 mC/m^2^ and 50.1 mC/m^2^, respectively, indicating increases of 32.6% and 50% compared with that of pristine PI. It is worth noting that the terp/PI-5 composite films demonstrate a lower residual polarization (P_r_) than that of pristine PI, which may arise from the tighter package in the PI chain with increasing terpolymer concentration. Furthermore, as the P(VDF-TrFE-CFE) concentration increased to 10%, the P_r_ of terp/PI-10 gradually increased. This can be attributed to the enhanced polarization loss of the terpolymer, which acts as a relaxor ferroelectric polymer. A reduced P_r_ is advantageous for enhancing the charging and discharging efficiency of terp/PI composite films. The energy density and charge–discharge efficiency, η, of the terp/PI composite films were calculated based on the P-E loop data; the results are shown in Figure 3b,c. To verify the accuracy of the data on energy density and charge–discharge efficiency, more than five samples of each terp/PI film with various terpolymer contents were measured and the detailed data with error bars are shown in Appendix A. Due to the higher Ps and lower Pr, under the same electric field conditions, the terp/PI composite film exhibited higher U_d_ and η. Accordingly, as shown in Figure 3b, the pristine PI exhibited an energy density of 6.05 J/cm^3^ and an efficiency of 68.5% at 500 MV/m. In contrast, terp/PI-1, terp/PI-5, and terp/PI-10 exhibited energy densities of 8.91 J/cm^3^, 9.67 J/cm^3^, and 9.29 J/cm^3^, representing increases of 47.3%, 59.8%, and 53.5%, respectively. Therefore, the terp/PI-5 film exhibited the highest energy density (9.67 J/cm^3^) and a relatively high charge–discharge efficiency (84.7%) among the terp/PI composite films. This may result from the increased dielectric constant and the highest breakdown field.

Subsequently, we employed the Weibull distribution equation to analyze the breakdown strength of the PI-based composite materials, which is expressed as follows:(1)$$P\left(E\right)=1-e^{\left[-{\left(\frac{E}{E_{b}}\right)}^{\beta }\right]}$$
where E is the measured breakdown strength under an alternating current (AC) field, E_b_ is the characteristic breakdown field at which 63.2% of the samples fail, β is a shape parameter that reflects the dispersion range of the measurement data, and P(E) represents the cumulative probability of failure for the tested samples [19]. As depicted in Figure 3d, the E_b_ values of terp/PI-1, terp/PI-5, and terp/PI-10 increased from 502.7 to 553.2, 604.7, and 557.8 MV/m, respectively, representing increases of 10.05%, 20.29%, and 10.96%, respectively, compared with that of the pristine PI film, indicating that the addition of the terpolymer enhances the breakdown strength of the terp/PI films.

The P-E loops of the terp/PI films were measured at 150 °C and the data are shown in Figure 4a. The data indicate that the breakdown field of terp/PI-5 achieved the highest value of 466.4 MV/m, which is 25.2% greater than that of the pristine PI film at 150 °C. Similarly, the E–P loops of the pristine PI film and terp/PI films at 200 MV/m (Figure 4b) show that the PI film exhibits a very high loss at high temperatures, which may increase the dielectric loss and result in thermal breakdown. In contrast, the terp/PI films exhibit a slender hysteresis loop, while maintaining a low Pr value. Afterward, the energy density and charge–discharge efficiency were determined based on the E–P loops at 150 °C (Appendix A). As shown in Figure 4b, the energy density of the terp/PI-5 film reached 5.06 J/cm^3^ at 475 MV/m, which is an 87.4% improvement compared to that of pristine PI (2.7 J/cm^3^) at 150 °C. As shown in Figure 4c, the charge–discharge efficiency is 95.6% at 200 MV/m for the terp/PI-5 film, which is 15.6% higher than that of pristine PI at the same electric field. These results suggest that the incorporation of the terpolymer in the PI substrates can successfully attenuate the electrical conductivity loss of the PI substrate at high temperatures and improve the energy density of the composite films. In addition, the data indicate that the charge–discharge efficiency of the terp/PI film significantly decreases when the electric field exceeds 300 MV/m at 150 °C. From previous publications, the discrepancy in dielectric constants between PVDF-based polymers and PI may cause the accumulation of electric fields at the interface between the electrode and dielectric material, leading to an enhanced injection of electrons under high temperature and high electric field conditions [39]. The breakdown field and charge–discharge efficiency of terp/PI films at high temperature could be reduced by this effect. Furthermore, the breakdown field of the terp/PI composite film was analyzed using Weibull distribution and the data are shown in Figure 4d. The data indicate that the breakdown field (E_b_) of the pristine PI sharply decreased to 372.4 MV/m at 150 °C. Meanwhile, the E_b_ of terp/PI-5 reached 466.4 MV/m, which is 25.2% higher than that of pristine PI film. This is consistent with the P-E loop data.

As a commonly used high T_g_ polymer, the presence of imide groups in the backbone of PI films results in excellent mechanical and thermal stability. However, the π-π stacking interactions between imide rings may reversely lead to charge conduction, resulting in a noticeable increase in the electrical conductivity of PI at high temperature, which may significantly increase the conduction loss of the material and reduce its efficiency [9,40]. Typically, to elucidate the mechanism behind the enhanced efficiency of terp/PI composite films, the electrical conductivity of the terp/PI composite films was measured at 150 °C using a Radiant Precision Multiferroic System; the data are shown in Figure 4e. The data were fitted using Schottky emission fitting software under a low electric field. The details of the conductive model are described in the Appendix A. Under a low electric field (<200 MV/m), the conduction mechanism of the pristine PI and terp/PI films obeyed the Schottky emission mechanism, as shown in Appendix A. Therefore, with increasing electric field, the conductive model changed from the Schottky emission model to the trap-limited hopping conduction model for both pristine PI and terp/PI films [25,41,42,43]. Based on previous research, this may be attributed to the trap sites formed by the terpolymer in the terp/PI composites. These trap sites may capture the electrons injected from the electrode to generate an internal electric field, in opposition to the applied electric field. Therefore, electron injection via Schottky emission is further suppressed [25]. At a higher electric field (E > 200 MV/m), the hopping conduction model was utilized to simulate the conductivity current of the terp/PI composites using Appendix A; the fitting results are shown by the dash line in Figure 4e. From the fitting results, the obtained jump distances, λ, for pristine PI and the terp/PI-1, terp/PI-5, and terp/PI-10 films were 1.89 nm, 1.79 nm, 1.47 nm, and 1.84 nm, respectively. Due to the shorter distance between the trap sites in the terp/PI films, the suppression of leakage current and reduction in conductance loss are more effective [42,44,45].

In the hopping conduction model, carriers must acquire energy that is equal to the trap depth to reach the conduction current. The energy required by the carrier to overcome the trap barrier is called the conductive activation energy. Therefore, the conductivity activation energy can reflect the trap depth of a material [18]. The change in electrical conductivity with temperature follows the Arrhenius model, as follows [25,44]:(2)$$\sigma \left(T\right)=\sigma_{0}\times e^{\left(-\frac{Aq}{Tk_{b}}\right)}$$
where σ(T) is the conductivity at different temperatures, σ_0_ is the prefactor, A is the conductive activation energy (eV), T is the temperature, k_b_ is the Boltzmann constant, and q is the electric charge. The conductivity activation energy can be obtained using the fitted slope of the conductivity as a function of the reciprocal of temperature. Figure 4f shows the fitted results obtained using the Arrhenius model for the conductivity activation energy of the terp/PI composite films at different temperatures under 200 MV/m. From the fitting results, the conductivity activation energies of pristine PI, terp/PI-1, terp/PI-5, and terp/PI-10 are 0.42 eV, 0.54 eV, 0.66 eV, and 0.52 eV, respectively. This means that the addition of the terpolymer to the terp/PI films results in higher activation energies, indicating an increased energy barrier. In addition, terp/PI-5 exhibited the highest conductivity activation energy, which may have resulted in a higher breakdown field and lower leakage current.

The thermally stimulated depolarization current (TSDC) is always used for evaluating the release of trapped charges during the heating process. The TSDC data can provide valuable insights into the energy levels and distribution of traps within the material [46]. As depicted in Figure 5a, the pristine PI film exhibited a single depolarization process, whereas the terp/PI-5 composite films displayed two depolarization processes. First, the low-temperature peak at approximately 50 °C for the terp/PI-5 composite film is associated with the dipole relaxation of the terpolymer. The high temperature depolarization of the terp/PI-5 composite at 171.52 °C is related to the charge release process in the trapped sites [47].

Compared to that of pristine PI films (151.52 °C), the depolarization temperature (T_P_) of the terp/PI-5 composite films shifted to a higher temperature peak of 171.52 °C, indicating that the addition of the terpolymer introduces deeper trap levels in the PI substrate [18,19,25]. According to the TSDC curve, the average trap depth (E_TSDC_) and trapped charge (Q_TSDC_) of the sample can be calculated using the full width at half maximum (FWHM) of the discharge peak; the equation can be expressed as follows [40,42]:(3)$$E_{TSDC}=\frac{2.47k_{b}{T_{P}}^{2}}{\Delta T}$$
(4)$$Q_{TSDC}=\left(\frac{60}{\upsilon }\right)\int_{T_{0}}^{T_{1}}I(T)dT$$
where k_b_ is the Boltzmann constant, T_P_ is the depolarization current, ΔT is the FWHM of the discharge current peak, ν is the heating rate (2 K/min), and T_0_ and T_1_ are the starting and ending temperatures of the depolarization current peaks, respectively. According to Equations (3) and (4), the calculated E_TSDC_ for PI and terp/PI-5 are 0.39 and 0.64 eV, respectively, and the Q_TSDC_ are 12.03 and 19.69 nC, respectively. The TSDC results indicate that adding terpolymer to the PI film can form deeper traps, as well as additional carrier traps, which can effectively reduce the conductance loss and eventually enhance the energy density and efficiency of the terp/PI composite [48].

Based on the experimental results, Multiwfn software was used to simulate the distribution of the surface electrostatic potential (ESP) of the terpolymer and PI film and to determine a more comprehensive profile for the formation of deep traps in terp/PI composites. The simulation results (Figure 5b and Appendix A) indicate that the terpolymer molecule exhibits a large percentage of the positive charge on its surface compared to the PI molecular structure [25,49]. This suggests that the terpolymer has a stronger affinity for electron capture as an effective trap site [50].

Subsequently, the band structures of the PI and terpolymer were analyzed using UV–Vis–NIR) and UV photoelectron spectroscopy (UPS). As shown in Appendix A, the calculated band gaps (E_g_) for the PI film and terpolymer film are 2.83 eV and 3.24 eV, respectively. Therefore, the ionization energies (ϕ_IP_) of both the PI and P(VDF-TrFE-CFE) films were calculated using the following equation [42]:(5)$$\phi_{IP}=hv-(E_{cutoff}-E_{homo})$$
where *hv* is the incident light energy (21.2 eV) from the UV–Vis equipment, E_cutoff_ is the transaction energy of secondary electrons, and E_homo_ represents the lowest energy required to bind electrons in the dielectric material [39,46]. The electron affinity E_a_ can be calculated based on the following equation [25]:(6)$$E_{a}=\phi_{IP}-E_{g}$$

The calculated results are presented in Figure 5c. From the data, the electron affinity energies of PI and P(VDF-TrFE-CFE) are −2.36 eV and −3.23 eV, respectively, which is consistent with the simulation results of the surface electrostatic potential distribution for both polymer structures. As shown in the schematic for the energy band structure illustrated in Figure 5d, the terpolymer material first captures excited electrons from the PI material due to its high electron affinity. This is attributed to the strong electrostatic attraction from the trap levels in the terpolymer film, which can effectively inhibit conductivity loss and enhance the energy density and charge–discharge efficiency of the terp/PI composites. The band structure of the interface of the terp/PI composite material is illustrated in Appendix A, where ΔEa represents the trap energy level (0.87 eV), which was calculated from the difference in electron affinities between these two materials. In general, the above results confirmed that a moderate amount of terpolymer can create energy traps in the PI film, which may significantly inhibit charge injection and transport and effectively reduce conduction losses at high temperatures in terp/PI composites [51].

In addition, the durability for the electric breakdown field of the Terp/PI-5 composite film was measured based on the E–P loop data at 150 °C. First, the energy density of Terp/PI-5 was tested under different holding periods. As shown by the E–P loop data in Appendix A, there were no significant decreases in either the energy density or charge–discharge efficiency of the sample after being stored at 150 °C for 90 min. Subsequently, the thermal stability of the nanocomposite film was measured by repeatedly charging and discharging the sample at 150 °C for 20,000 cycles. As shown in Figure 6a, the energy density of terp/PI-5 remains at 1.17 J/cm^3^ after 20,000 charge–discharge cycles at 150 °C at 200 MV/m, which is a decrease of 7.1% compared to the value (1.26 J/cm^3^) before the measurement. Moreover, a charge–discharge efficiency of 93.9% was maintained after the measurement. The results indicate that the terp/PI-5 composite exhibits excellent durability and reliability at high temperature (150 °C). Furthermore, Appendix A shows a summary of the energy density of the terp/PI film compared to recently published results for PI-based films at high temperature (150 °C) [13,19,27,33,48,52,53,54,55,56]. The results indicate that the terp/PI film exhibits superior energy density and electric breakdown field at high temperature. Therefore, terp/PI composite films show great promise for high energy density capacitor applications at high temperatures.

## 4. Conclusions

In this investigation, a small amount of relaxor ferroelectric terpolymer (P(VDF-TrFE-CFE)) has been incorporated into polyimide (PI) substrates to create all-organic terpolymer/PI (terp/PI) composite films for high-energy density capacitor applications. The permittivities of the terp/PI-1, terp/PI-5, and terp/PI-10 reached 3.51, 3.81, and 4.27, respectively, at 1 kHz, which are 9.3%, 18.7%, and 33% greater than that of pristine PI (3.21). The terp/PI-5 film exhibited the highest Young’s modulus, herein, among terp/PI composites. Therefore, terp/PI-5 film was focused on to investigate high-energy density capacitor application. The terp/PI-5 film exhibited the highest energy density (9.67 J/cm^3^) and a relatively high charge–discharge efficiency (84.7%). The energy density of the terp/PI-5 film was increased by 59.8% compared to that of the pristine PI film. This may result from the increased dielectric constant and the highest breakdown field. As depicted in Figure 4b, the energy density of the terp/PI-5 film reached 5.06 J/cm^3^ at 475 MV/m at 150 °C, which is an 87.4% improvement compared to that of pristine PI (2.7 J/cm^3^). The TSDC results and band structure analysis revealed the presence of deeper traps in the terp/PI composites, contributing to the suppression of leakage current and improved charge–discharge efficiency. The terp/PI composite films exhibited remarkable stability under high temperatures, making them promising candidates for applications in high-performance capacitors. This study creates a new approach using a small amount of relaxor ferroelectric polymer doping in PI substrates to produce capacitor materials with high-energy density and improved breakdown field at high temperatures. 

## Figures and Tables

**Figure 1 polymers-16-01138-f001:**
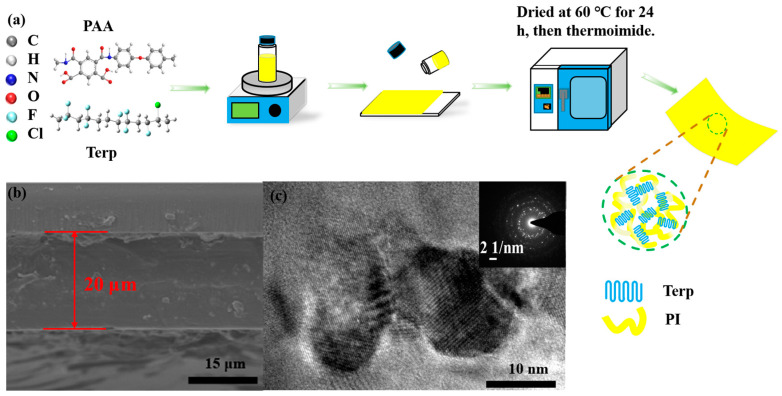
(**a**) Schematic diagram of the fabrication process of the Terp/PI nanocomposite film, (**b**) scanning electron microscopy image of the cross-sectional surface morphology, and (**c**) transmission electron microscopy image of the Terp/PI-5 film.

**Figure 2 polymers-16-01138-f002:**
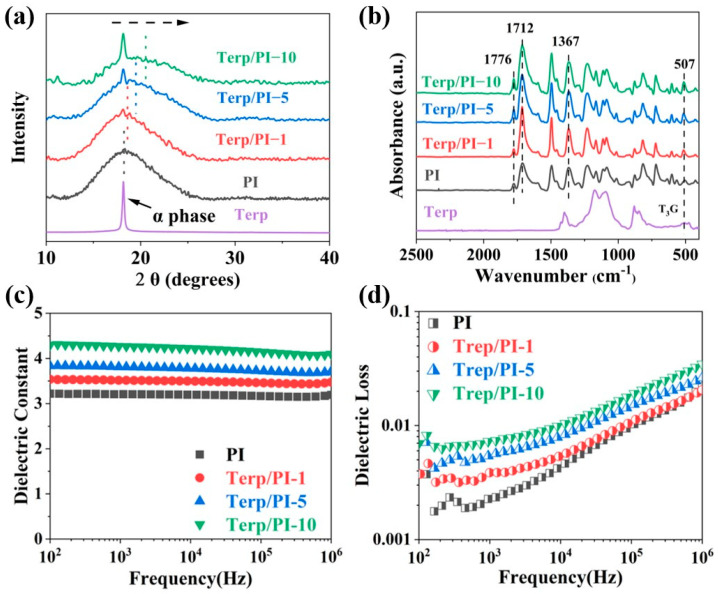
(**a**) X-ray diffraction (XRD) patterns, the arrow represents the direction of movement of the amorphous peak. (**b**) Fourier transform infrared (FTIR) spectra of pristine PI and terp/PI composite films. (**c**) Dielectric constant and (**d**) dielectric loss of Terp/PI composite films with various terpolymer contents at different frequencies.

**Figure 3 polymers-16-01138-f003:**
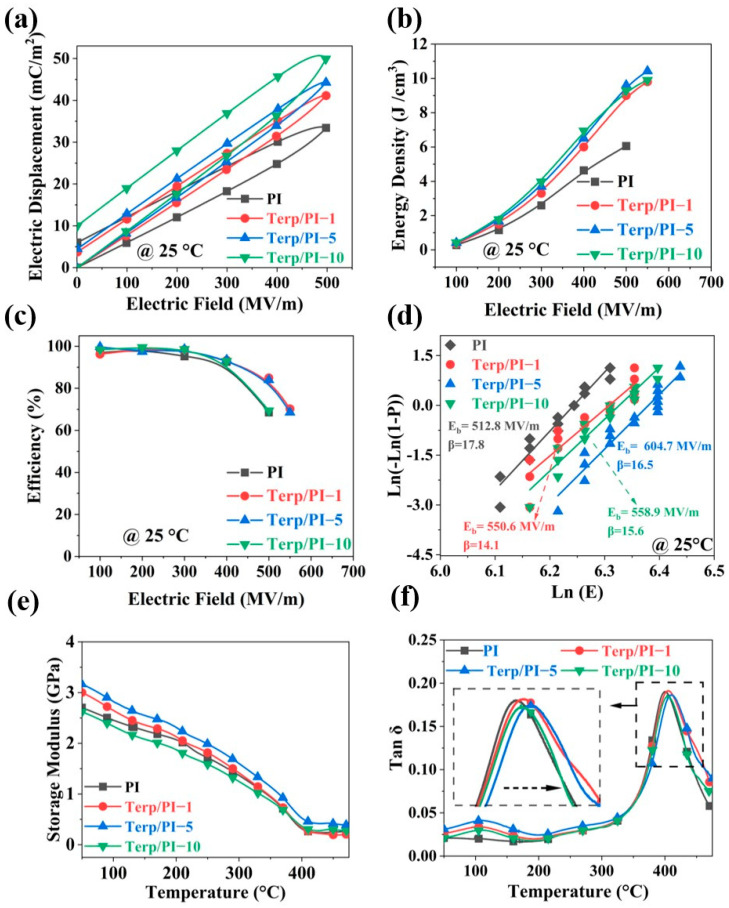
(**a**) P−E loops, (**b**) energy density, (**c**) charge−discharge efficiency, and (**d**) Weibull distribution for the electric breakdown field of the terp/PI films with different contents at room temperature, the arrows represent the specific result of Weibull distribution, in which arrows represent the breakdown fields and shape parameters of the terp/PI composite films with various terpolymer contents. (**e**) Young’s modulus and (**f**) mechanical loss factor measured using DMA.

**Figure 4 polymers-16-01138-f004:**
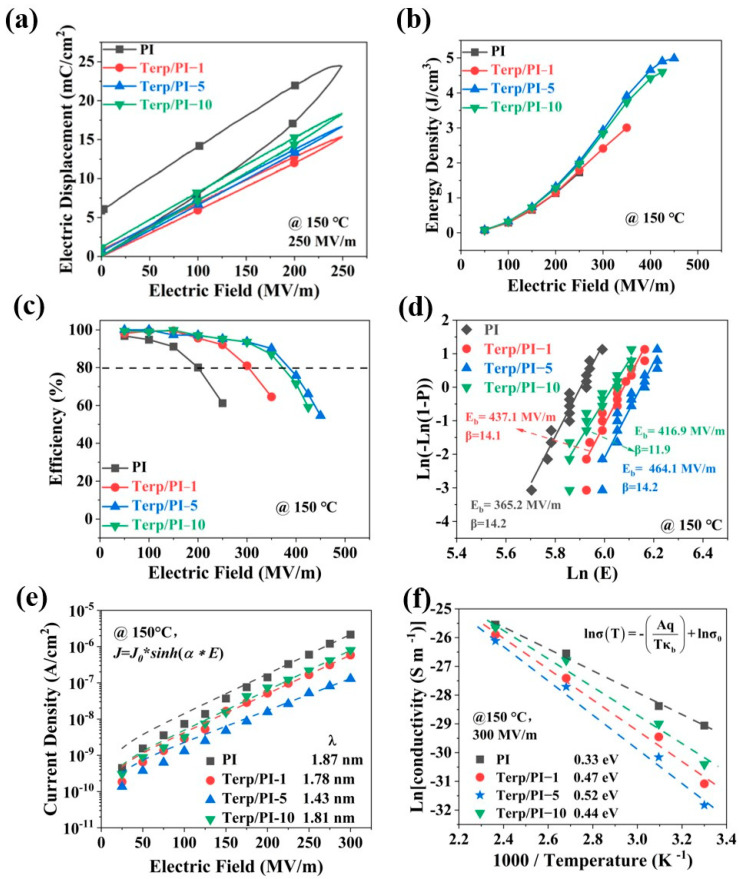
(**a**) P−E loop at 200 MV/m, (**b**) energy density and (**c**) charge−discharge efficiency, and (**d**) Weibull distribution for the electric breakdown field of the terp/PI composite films measured at 150 °C. (**e**) Fitted results of the current density as a function of the applied electric field at 150 °C for the pristine PI and terp/PI films using the hopping conduction model. (**f**) Fitted results for the conductivity as a function of reciprocal of temperature for the pristine PI and terp/PI composites measured at 200 MV m^−1^ using the Arrhenius function.

**Figure 5 polymers-16-01138-f005:**
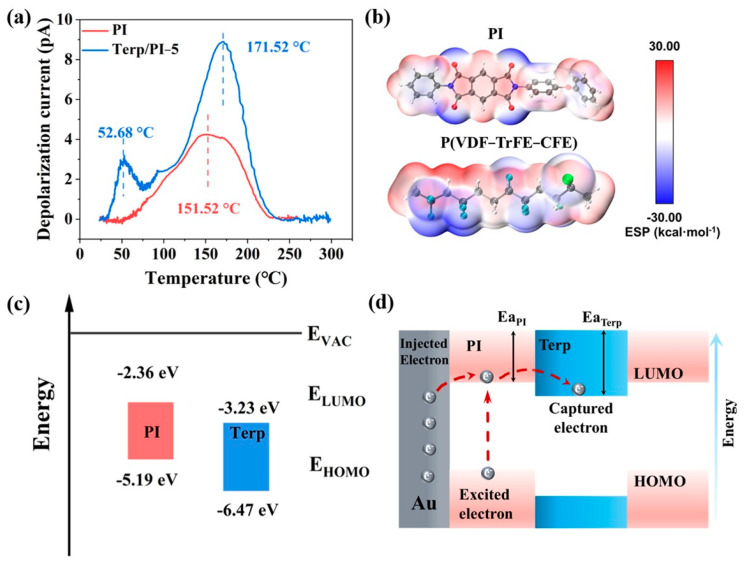
(**a**) Thermal stimulated discharge current curves of the PI and terp/PI-5 films. (**b**) Simulation results of the electrostatic potential distribution on the molecular surface of PI and P(VDF−TrFE−CFE). (**c**) Schematic drawing of the energy band diagram for the PI and P(VDF−TrFE−CFE) films. (**d**) Schematic band diagram for the interface between the terpolymer and the PI substrate in the composite.

**Figure 6 polymers-16-01138-f006:**
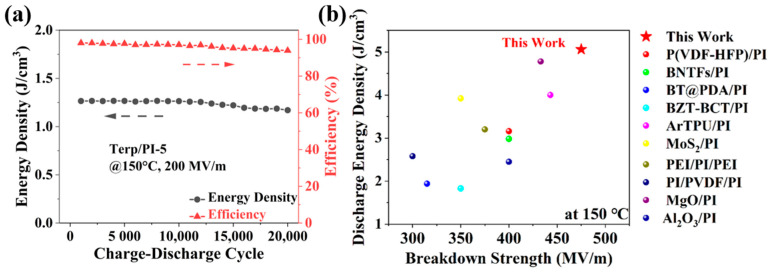
(**a**) Reliability test for the Terp/PI-5 composite film at 200 MV/m and 150 °C. (**b**) Comparison of the energy storage properties of the Terp/PI composites and other PI-based high-temperature dielectrics at 150 °C.

## Data Availability

Data are available by request.

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
