# Peer review of "High Energy Density in All-Organic Polyimide-Based Composite Film by Doping of Polyvinylidene Fluoride-Based Relaxor Ferroelectrics"

_polymers, 2024, doi:10.3390/polym16081138_

Round 1
Reviewer 1 Report
Comments and Suggestions for Authors
Authors produce all-organic P(VDF-TrFE-CFE) terpolymer/PI (terp/PI) com-12 posite films by incorporating a small amount of terpolymer into PI substrates for high energy density capacitor applications.
In known works, due to the relatively smaller breakdown field of terpolymer films, their use in capacitors with high energy density is limited, especially at high temperatures.
Authors indicate that in previous studies have shown that by introducing polar nanoregions (PNRs) into relaxor ferroelectrics, relaxor ferroelectrics can exhibit significantly improved saturation polarization (Ps) and increased energy density.
In this study, a small amount of terpolymer was incorporated into PI substrates to fabricate an all-organic terp/PI polymer (terp/PI) composite film using a solution casting method.
The second idea of the work is to show that the introduction of a terpolymer can create deep traps that suppress the leakage current and improve the electric breakdown field of the composite film.
Methodologically, the work was done correctly. All measurements taken fully correspond to the goals set in the work and fully characterize the research material.
The conclusions of the article are fully consistent with the experimental data presented. All the main goals of the work were achieved through experiments to investigate the reason of the enhanced breakdown field of the terp/PI composites, the mechanical properties of the terp/PI films were assessed using DMA.
Dynamic mechanical analysis (DMA) is one of the important methods that is used to study the dependence of the mechanical and viscoelastic properties of materials on temperature, time and frequency under the influence of periodic loads.
All references given in the work fully disclose previously obtained data in the world literature.
In all figures in the article where experimental data are presented, it is necessary to show the scatter whiskers (error) when obtaining this data.
Your article is of interest to readers of the magazine "Polymers". I do not have any comments on the content and format of the article.
Author Response
Point to Point Response
Reviewer #1:
Authors produce all-organic P(VDF-TrFE-CFE) terpolymer/PI (terp/PI) composite films by incorporating a small amount of terpolymer into PI substrates for high energy density capacitor applications. In known works, due to the relatively smaller breakdown field of terpolymer films, their use in capacitors with high energy density is limited, especially at high temperatures. Authors indicate that in previous studies have shown that by introducing polar nanoregions (PNRs) into relaxor ferroelectrics, relaxor ferroelectrics can exhibit significantly improved saturation polarization (Ps) and increased energy density. In this study, a small amount of terpolymer was incorporated into PI substrates to fabricate an all-organic terp/PI polymer (terp/PI) composite film using a solution casting method. The second idea of the work is to show that the introduction of a terpolymer can create deep traps that suppress the leakage current and improve the electric breakdown field of the composite film. Methodologically, the work was done correctly. All measurements taken fully correspond to the goals set in the work and fully characterize the research material. The conclusions of the article are fully consistent with the experimental data presented. All the main goals of the work were achieved through experiments to investigate the reason of the enhanced breakdown field of the terp/PI composites, the mechanical properties of the terp/PI films were assessed using DMA. Dynamic mechanical analysis (DMA) is one of the important methods that is used to study the dependence of the mechanical and viscoelastic properties of materials on temperature, time and frequency under the influence of periodic loads. All references given in the work fully disclose previously obtained data in the world literature. In all figures in the article where experimental data are presented, it is necessary to show the scatter whiskers (error) when obtaining this data.
Your article is of interest to readers of the magazine "Polymers". I do not have any comments on the content and format of the article.
Answer: Thank you for your thoughtful feedback. We have re-measured the energy density and charge-discharge efficiency of the terp/PI composites with various terpolymer contents and the following explanation and data were added to the revised manuscript and electronic supplementary information (ESI).
Pg.14, Para.1: To verify the accuracy of the data on energy density and charge-discharge efficiency, more than 5 samples of each terp/PI film with various terpolymer contents were measured and the detailed data with error bars are shown in Figure S13a and b (ESI).

Reviewer 2 Report
Comments and Suggestions for Authors
In this paper, all-organic PVDF based terpolymer / PI composite films were produced for high energy density capacitor applications. This paper is designed logically, all results are presented with high quality, and discussions well addressed. The knowledge obtained from this study will help with high e capacitor applications. I would suggest acceptance after a few issues:
1. Fig 2d, the increase of e'' under high frequencies, could be attributed to bad conduct. Please make sure the fixture that direct contact with gold coated samples are coated/plated with gold as well.
2. For dielectric tests, please show more information at lower frequencies (0.1 Hz-1000 Hz). I understand there are huge conduction loss at low frequencies, but still should include in the curves.
3. Fig 2d, please change dielectric loss to log scale, expecially if any conduction peaks exist.
4. Drying at 60 °C for 24h is way not enough. Should dry at high temperature (at least 100 °C). These moisture will create huge huge huge loss in P-E loops due to charge injection (AC conduction loss).
5. Fig 3d, Weibull analysis please test more samples. I understand that home made samples are limited by sample size, but 10+ samples can reveal more accurate information.
6. Please include bipolar P-E loops in supplemantal information.
7. The PVDF-PI composite will create high charge injection, especially high temperature and high field. I believe authors have found this, e.g., 300MV/m or higher at 100-150 °C. This is because PVDF-PI dielectric difference, which cause way higher electric field condensed regions. This paper on charge injection worth further discussed and should be cited: https://doi.org/10.1016/j.ensm.2021.12.009
8. The discussion and simulation on Schottky is great. I like it.
Author Response
Point to Point Response to Reviewer #2
Reviewer #2:
- Fig 2d, the increase of ε'' under high frequencies, could be attributed to bad conduct. Please make sure the fixture that direct contact with gold coated samples are coated/plated with gold as well.
Answer: We sincerely appreciate the valuable comments. The following explanation has been added to the revise manuscript.
Pg. 13, Para. 2: During this measurement, a 200 g metal piece was placed on top of the holder of an HP LCR meter to ensure the good contact between the holder and the composite sample. According to the literature, the high dielectric loss in the terp/PI-10 sample could be caused by the dipole relaxation in the terpolymer.
- For dielectric tests, please show more information at lower frequencies (0.1 Hz-1000 Hz). I understand there are huge conduction loss at low frequencies, but still should include in the curves.
Answer: Thank you for your thoughtful feedback. We remeasured the permittivity of the terp/PI composite from 100 Hz and the result are shown in Figures 2c and 2d in the revised manuscript. Due the limitation of HP LCR meter, the dielectric constant data below 100 Hz were not measured, which may show a large error with current equipment.
Pg. 13, Para. 2: In addition, the data also indicate that the dielectric loss was slightly high at low frequencies, which could be due to the inconsistency of the HP LCR meter at low frequencies (< 1 kHz)
- Fig 2d, please change dielectric loss to log scale, expecially if any conduction peaks exist.
Answer: Thank you for your thoughtful feedback. We have changed the dielectric loss to log scale and the result is show in Figure 2d in the revised manuscript.
- Drying at 60 °C for 24h is way not enough. Should dry at high temperature (at least 100 °C). These moisture will create huge huge huge loss in P-E loops due to charge injection (AC conduction loss).
Answer: We sincerely appreciate the valuable comments. The following explanation has been added to the revised manuscript.
Pg. 6, Para. 2: The obtained terp/PAA film was heated at 200 °C for 12 h to remove the residual solvent and moisture. This step is also beneficial to the thermalimide process of the PAA film. Afterwards, the terp/PAA film was heated at 250 °C and 300 °C for 2 h to continue the thermalimide process and obtain the terp/PI film.
- Fig 3d, Weibull analysis please test more samples. I understand that home made samples are limited by sample size, but 10+ samples can reveal more accurate information.
Answer: Thank you for your thoughtful feedback. The Weibull distribution was reanalyzed with more than 15+ samples for the terp/PI film with different contents and the results are shown in Figure 3d and Figure 4d in the revised manuscript.
- Please include bipolar P-E loops in supplemantal information.
Answer: Thank you for your suggestion. The bipolar P-E loops and the results are shown in Figure S14 and Figure S15 in the revised supporting information.
- The PVDF-PI composite will create high charge injection, especially high temperature and high field. I believe authors have found this, e.g., 300MV/m or higher at 100-150 °C. This is because PVDF-PI dielectric difference, which cause way higher electric field condensed regions. This paper on charge injection worth further discussed and should be cited: https://doi.org/10.1016/j.ensm.2021.12.009.
Answer: We sincerely appreciate the valuable comments. The suggested paper thoroughly investigates the electric field accumulation between the electrode and dielectric caused by differences in dielectric properties. Therefore, this paper was cited and the following explanation has been added to the revised manuscript.
Pg. 15, Para. 1: In addition, the data indicate that the charge-discharge efficiency of the terp/PI film significantly decreases when the electric field exceeds 300 MV/m at 150°C. From previous publications, the discrepancy in dielectric constants between PVDF polymers and PI may cause the accumulation of electric fields at the interface between them, leading to an enhanced injection of electrons under high temperature and high electric field conditions [38]. The breakdown field and charge-discharge efficiency of terp/PI films at high temperature could be reduced by this effect.

Reviewer 3 Report
Comments and Suggestions for Authors
It is an interesting study on the fabrication of all-organic polyimide-based composite films doped with P(VDF-TrFE-CFE) relaxor ferroelectrics for high-energy density capacitor applications. The work is well done, and I recommend its publication once the authors address the following queries:
1. The authors in the introductory paragraph state “However, due to the relatively lower breakdown field of terpolymer films, their use in high-energy density capacitors is limited”. What is the breakdown field of the terpolymer?
2. In section 2.1, “1.5 g of polyamide acid solution (PAA) was added to 7.0 ml of N’N-dimethylacetamide (DMF) …”. Here is a typographical error. Did the authors utilize DMF or DMA? Additionally, please include specifics regarding the temperature conditions.
3. Was P(VDF-TrFE-CFE) concentration calculated relative to the entire solution or just to PI/P(VDF-TrFE-CFE) mixture in the solution?
4. Why does the Terp/PI-10 exhibit the highest dielectric constant despite the evident phase segregation, as illustrated in Figure S2? The authors are requested to provide a valid explanation for the observed outcome.
5. The authors should review the labeling of figures in the Supporting Information and ensure accurate descriptions in the main manuscript. Several instances of misassignment have been noted, such as the absence of Figures S13 and S14, despite their indication in the main manuscript.
6. It is recommended that the authors incorporate actual images of the fabricated composite films.
Author Response
Point to Point Response to Reviewer #03
Reviewer #3:
Comments:
- The authors in the introductory paragraph state “However, due to the relatively lower breakdown field of terpolymer films, their use in high-energy density capacitors is limited”. What is the breakdown field of the terpolymer?
Answer: Thank you for your thoughtful feedback. From previous publications, the electric breakdown field of terpolymer at room temperature is usually lower than 350 MV/m. We have corrected the description in the text manuscript and highlighted it:
Pg. 5, Para .2: However, due to the relatively low Tg and breakdown field (< 350 MV/m) of terpolymer films, their use in high-energy density capacitors is limited, especially at high temperatures [26-30].
- In section 2.1, “1.5 g of polyamide acid solution (PAA) was added to 7.0 ml of N’N-dimethylacetamide (DMF)…”. Here is a typographical error. Did the authors utilize DMF or DMA? Additionally, please include specifics regarding the temperature conditions.
Answer: Thank you for your thoughtful feedback. The point you mentioned is not a typographical error; the solvent employed in the experiment is N’N-dimethylacetamide (DMF). The temperature for the fabrication of the PAA/DMF solution was added to the revised manuscript.
Pg. 6, Para. 2: First, 1.5 g of polyamide acid solution (PAA) (15 wt.%, Saint Marvel, Run Chuan Plastic Materials Co., Changzhou, China) was added to 7.0 ml of N’N-dimethylacetamide (DMF) solvent in a clear glass bottle, and stirred for 4 hours at room temperature until it was completely dissolved.
Pg. 6, Para. 2: The mixture was sonicated for 2 h in a water bath at room temperature and then magnetically stirred for an additional 4 h until it was completely dissolved.
- Was P(VDF-TrFE-CFE) concentration calculated relative to the entire solution or just to PI/P(VDF-TrFE-CFE) mixture in the solution?
Answer: Thank you for your thoughtful feedback. The P(VDF-TrFE-CFE) concentration is calculated based on the mass of PAA powders. The following explanation has been added to the revised manuscript.
Pg. 6, Para. 2: Next, based on the mass of PAA powders, 1 wt%, 5 wt%, and 10 wt% P(VDF-TrFE-CFE) powders (59.2/33.6/7.2 mol%, Piezo Tech. Inc., France) were weighed
- Why does the Terp/PI-10 exhibit the highest dielectric constant despite the evident phase segregation, as illustrated in Figure S2? The authors are requested to provide a valid explanation for the observed outcome.
Answer: Thank you for your thoughtful feedback.
Pg. 12, Para. 3: From previous publications, it is known that the dielectric constant of the composite is proportional to the dielectric constants of the substrate and the dopant. Therefore, the terp/PI-10 exhibits the highest dielectric constant of all terp/PI composites (Figure S5). However, the molecular chain structure of terpolymer differs significantly from that of PI, and their physical mixing can result in phase separation between them when there is a high content of terpolymer. This may reduce the breakdown field of terp/PI composites with a high content of terpolymer. Therefore, our focus was on terp/PI-5 composite film for the study of high energy density capacitors.
- The authors should review the labeling of figures in the Supporting Information and ensure accurate descriptions in the main manuscript. Several instances of misassignment have been noted, such as the absence of Figures S13 and S14, despite their indication in the main manuscript.
Answer: Thank you for your thoughtful feedback. We have corrected all these misassignments in the revised manuscript.
- It is recommended that the authors incorporate actual images of the fabricated composite films.
Answer: Thank you for your thoughtful feedback. We have provided actual images of the fabricated composite films and the results are shown in Figure S1(e) in the supporting information. The following explanation has been added to the revise manuscript.
Pg. 9, Para. 2: the image of the fabricated terp/PI-5 film are shown in Figure S1(e) (ESI)
